# Impact of Nasal Swabs on Empiric Treatment of Respiratory Tract Infections (INSERT-RTI)

**DOI:** 10.3390/pharmacy8020101

**Published:** 2020-06-11

**Authors:** Vanessa Huffman, Diana Carolina Andrade, Jared Ham, Kyle Brown, Leonid Melnitsky, Alejandro Lopez Cohen, Jayesh Parmar

**Affiliations:** 1Department of Pharmacy, Memorial Hospital West, Pembroke Pines, FL 33028, USA; JHam@mhs.net (J.H.); KyBrown@mhs.net (K.B.); 2Department of Emergency Medicine, Memorial Hospital West, Pembroke Pines, FL 33028, USA; LMelnitsky@mhs.net; 3Department of Graduate Medical Education, Memorial Hospital West, Pembroke Pines, FL 33028, USA; alopezcohen@mhs.net; 4Department of Clinical and Administrative Sciences, Larkin College of Pharmacy, Miami, FL 33169, USA; JParmar@ULarkin.org

**Keywords:** antimicrobial stewardship, pneumonia, methicillin-resistant *Staphylococcus aureus*, vancomycin, nasal swab

## Abstract

Methicillin-resistant *Staphylococcus*
*aureus* (MRSA) polymerase-chain-reaction nasal swabs (PCRNS) are a rapid diagnostic tool with a high negative predictive value. A PCRNS plus education “bundle” was implemented to inform clinicians on the utility of PCRNS for anti-MRSA therapy de-escalation in respiratory tract infections (RTI). The study included patients started on vancomycin with a PCRNS order three months before and after bundle implementation. The primary objective was the difference in duration of anti-MRSA therapy (DOT) for RTI. Secondary objectives included hospital length of stay (LOS), anti-MRSA therapy reinitiation, 30-day readmission, in-hospital mortality, and cost. We analyzed 62 of 110 patients screened, 20 in the preintervention and 42 in the postintervention arms. Mean DOT decreased after bundle implementation by 30.3 h (*p* = 0.039); mean DOT for patients with a negative PCRNS decreased by 39.7 h (*p* = 0.014). Median cost was lower after intervention [USD$51.69 versus USD$75.30 (*p* < 0.01)]. No significant difference in LOS, mortality, or readmission existed. The bundle implementation decreased vancomycin therapy and cost without negatively impacting patient outcomes.

## 1. Introduction

The Center for Disease Control and Prevention recognizes Methicillin-resistant *Staphylococcus aureus* (MRSA) as a “serious threat” within the community and hospital settings [1]. MRSA, a common etiology of pneumonia, is responsible for high rates of morbidity, mortality, antimicrobial resistance, and healthcare costs, however hospitals may be overusing anti-MRSA therapies based on documented MRSA infection rates [2,3,4]. Current guidelines for the treatment of MRSA respiratory tract infections (RTI) recommend rapid initiation of adequate antibiotic therapy with vancomycin or linezolid. These two antibiotics constitute the treatment of choice for MRSA bacterial infections, but antimicrobial resistance has been steadily on the rise, demanding new strategies to preserve the utility of these agents for the future [5,6,7]. In addition, therapy with these antimicrobials are not without their own risks including acute kidney injury (AKI), nephropathy, and/or serotonin syndrome [5,8]. In suspected RTI, common culture-identifying tests, including sputum cultures and bronchi-alveolar lavage (BAL), are recommended to guide therapy. These methods could be inconclusive due to insufficient or contaminated samples, respiratory tract bacterial colonization, or sample collection after antibiotic administration [6]. BAL fluid cultures are sensitive and specific for bacterial pneumonia. However the procedure is invasive and not always feasible to conduct [9].

Multiple studies have identified MRSA polymerase-chain-reaction nasal swabs (PCRNS) to have high negative predictive values (NPV) (94–99%) and low positive predictive values (PPV) (35–37%) with a high sensitivity and specificity [9,10,11,12,13,14,15]. The high negative predictive value implies that pneumonia due to MRSA is highly unlikely. Poor positive predictive values, on the other hand, suggests that a positive PCRNS does not confirm MRSA pneumonia and that initiation of MRSA targeted therapy is not indicated. The Infectious Disease Society of America (IDSA) Hospital-Acquired and Ventilator-Associated Pneumonia (HAP/VAP) guidelines state only 30% of RTI are due to MRSA in patients with positive PCRNS [6]. This evidence has introduced MRSA PCRNS as an effective tool to safely de-escalate antibiotic therapy [16,17,18,19,20].

Studies have implemented the use of MRSA PCRNS with pharmacist-led antimicrobial stewardship programs (ASP) to reduce the duration of therapy (DOT), cost, adverse events, hospital length of stay (LOS), and improve clinical outcomes [21,22,23,24]. The 2019 IDSA and American Thoracic Society (IDSA/ATS) Community-Acquired Pneumonia (CAP) guidelines were updated to suggest using PCRNS as a diagnostic tool for de-escalation of anti-MRSA therapy [7].

This study describes a novel multidisciplinary approach to the implementation of an ASP-PCRNS educational program to evaluate the impact in reducing anti-MRSA therapy for RTI, LOS, hospital readmission, and cost.

## 2. Materials and Methods

This is a quasi-experimental study comparing the duration of anti-MRSA therapy in patients diagnosed with RTI before and after the implementation of an educational program on the usage of MRSA PCRNS for de-escalation of anti-MRSA therapy. The study was conducted at a 502-bed, nonprofit community teaching hospital. The institution IRB deemed the study exempt as this study was a quality improvement pilot. The preintervention and postintervention arms were between 1 June 2019 and 30 August 2019, and 1 October 2019 to 1 January 2020, respectively. The month of September 2019 was a washout period to allow for education and process implementation. Patients met inclusion criteria if they were 18 years of age or older and started on vancomycin or linezolid for a suspected RTI with a MRSA PCRNS ordered. Patients were excluded if they presented with febrile neutropenia, received decolonization therapy (nasal mupirocin) prior to the PCRNS, had concomitant infections besides RTI, or had positive MRSA cultures. A prospective patient list was generated and reviewed daily including weekends using an in-house antibiotic tracking system that recorded antimicrobial agents and their indications. Patients identified on vancomycin or linezolid with RTI as the only documented indication by the tracking system were then screened for inclusion in the study. The primary objective evaluated if incorporation of an ASP-PCRNS bundle would reduce the anti-MRSA agent DOT for RTI. The secondary objectives evaluated LOS, reinitiation of anti-MRSA therapy, in-hospital mortality, 30-day RTI-related readmission, and cost of therapy for RTI.

### 2.1. ASP-PCRNS Bundle

The study implemented an ordering and surveillance process for pharmacists, medical residents, and physicians, which, paired with the interdisciplinary education process, was referred to as the “ASP-PCRNS bundle”.

Healthcare providers were educated to order a MRSA PCRNS for all patients who had met criteria and started on vancomycin or linezolid for an RTI, and to de-escalate anti-MRSA therapy if a negative PCRNS resulted. The pharmacists’ education on MRSA PCRNS utility was conducted via mandatory huddles and presentation handouts. Per protocol, pharmacists were permitted to order and review PCRNS independently. However, provider approval was required to discontinue anti-MRSA therapy. Pharmacists were trained to document a standardized written intervention in the patient’s chart and to contact the prescriber once the results were negative. An additional PCRNS results comment was included in the institution’s standardized pharmacokinetic notes to aid with bundle compliance (Appendix A).

Physician education was completed via departmental meetings, posted algorithms, and individual sessions by the emergency medicine physician champion (Appendix B). Admitting providers and leading users of anti-MRSA therapy for RTI were specifically targeted for education by the physician champion who provided either telephone or in-person education regarding the initiative. The medical residents were educated by the medical resident champion via grand rounds and critically appraised topic discussions. Educational materials included, but were not limited to, literature reviews, PowerPoint presentations, and workflow diagrams. ASP–PCRNS bundle educational materials are included in the supplemental appendix (Appendix B).

### 2.2. Data Collection

Patient baseline data was measured via multiple variables including age, body mass index (BMI), creatinine clearance, Charlson Comorbidity Index (CCI), and in-hospital use of concomitant nephrotoxins. Nephrotoxins were defined as any medication present on the American Family Physician table of drugs associated with nephrotoxicity [25]. Baseline health function and comorbidities were assessed using the CCI to predict a one-year mortality risk based on weighted composite score on comorbidities in categories such as cardiovascular, pulmonary, neurologic, renal, hepatic, gastrointestinal, and neoplastic. Scores were categorized into three grades: mild scores (1–2), moderate scores (3–4), or severe (≥5). Anti-MRSA therapy and MRSA PCRNS results were also collected for analysis. Data from the EMR were collected from the study period and entered by two primary researchers into the REDCap v9.7.2 software (https://projectredcap.org/) and reviewed for accuracy by study team members [26,27]. Protocol compliance was defined as PCRNS ordering within 36 h of anti-MRSA therapy initiation and de-escalation compliance was defined as discontinuation of anti-MRSA therapy within 36 h of PCRNS.

Cost of therapy was calculated by adding the costs of the MRSA PCRNS (USD $20), vancomycin levels (USD $13), and the daily medication cost (USD $9/day for vancomycin or USD $84/day for linezolid) with separate analyses being planned for each medication due to significant variance in costs. The cost of the PCRNS and serum level were provided by the laboratory department. The drug cost is representative of the average wholesale price per Lexicomp^®^ at the time of IRB submission.

### 2.3. Statistical Analysis

Based on an estimated standard deviation of 48 h and a two-sided alpha of 0.05, we estimated in an a priori calculation that a sample of 63 patients in each group—in total 126—was required in order to detect a 24 h difference in DOT between groups with a power of 80%. The primary outcome and additional continuous data were analyzed using the two-sample t-test or the Wilcoxon Signed Rank Test. Noncontinuous data was analyzed using nonparametric tests (Chi-Square) comparing medians of the pre and post groups. A regression analysis was conducted to evaluate the association of CCI and DOT. Subgroup analyses were conducted via the same methodology unless the data was non-normal or the sample size was seven or less, in which case the Mann-Whitney U test was used for analysis.

## 3. Results

### 3.1. Inclusion and Exclusion

A total of 253 patients were identified for screening, of which 143 patients were excluded without further screening, with documented concomitant infections, lack of anti-MRSA therapy on admission, and a lack of RTI diagnosis being the most common reasons for exclusion. Of the remaining 110 patients who were identified as having a diagnosis of RTI without other exclusion factors, 48 patients in total were excluded due to a lack of PCRNS order or a positive MRSA culture identified in Figure 1. Specifically, 10 of those patients lacking PCRNS were in the postintervention group, representing an ASP-PCRNS bundle compliance of 81% (42/52 patients).

### 3.2. Baseline Characteristics

Baseline characteristics were similar among both groups (Table 1). The average patient among both groups was approximately 65 years of age with a BMI of 26 kg/m^2^. There were more males in the post group, 57%, compared to the pre group, 50%. Majority of the patients were Caucasian, 65% in the pre group and 62% in the post group. Approximately 30% of the population was African American within each group. The post group identified one patient within more than one race or unknown/not reported. All patients in the pre group utilized concomitant nephrotoxic medications, whereas all patients but one in the post group had concomitant nephrotoxins. Among both groups the average patient scored severe in CCI. Creatinine clearance in both groups averaged greater than 60 mL/min, though patients in the post group had a numerically lower creatinine clearance as compared to the pre group. All patients were treated using vancomycin. No patient on linezolid met inclusion criteria.

In the preintervention group (*n* = 20), 18 patients had a negative PCRNS (90%) and 2 had a positive PCRNS. In the postintervention group (*n* = 42), 37 patients had a negative PCRNS (88%) and five patients had a positive PCRNS.

### 3.3. Primary and Secondary Outcomes

The overall DOT in all patients with a PCRNS was reduced at 59.7 h in the post group as compared to 90.7 h in the pre group (*p* = 0.039) (Table 2). In the post group 65% (24/37) of patients with a negative PCRNS were successfully de-escalated from anti-MRSA therapy within 36 h of PCRNS order, as compared to only 22% (4/18) in the pre group (Figure 2). Patients’ LOS increased in the post group by seven days compared to the pre group but was not statistically significant. In-hospital mortality was 10% in both groups (*p* = 0.953) and anti-MRSA therapy was restarted in 5% of patients in both groups (*p* = 0.97). Median cost was significantly lower after bundle implementation [USD$51.69 versus USD$75.30] (*p* < 0.01)].

Of those in the post group, 76% (32/42) of PCRNS were ordered within 36 h and 95% (40/42) were within 48 h. At the institution, on average, PCRNS results were available 20.2 h after the PCRNS order was placed. A subgroup analysis of patients who have received PCRNS > 36 h after vancomycin initiation was conducted. The median time to PCR within the 10 patients in the post group who did not receive PCNRS within 36 h of anti-MRSA therapy initiation was 44 h. Of the subgroup of 10, eight had a PCRNS within 48 h. The two patients who did not receive PCRNS within 48 h appeared to not have been acted upon as they received treatment courses of 5.6 and 8.75 days of therapy.

The DOT, cost, and LOS were analyzed among demographic subgroups including gender, age (<65 years old or ≥65 years old), ethnicity (Caucasian versus non-Caucasian), and BMI (<30 kg/m^2^ or ≥30 kg/m^2^) to assess for potential differences among groups. Duration of therapy was statistically significantly shorter in the post group among female patients, non-Caucasian patients, and patients with BMI less than 30 kg/m^2^. All other parameters did not meet statistical significance though each population had a numerically lower duration of therapy in the post group than in the pre group. Likewise, cost was overall reduced among all subgroups in the post group compared to those in the pre group. However only in patients with a BMI less than 30 kg/m^2^ was there a statistically significant reduction in cost: USD$80.62 in the pre group versus USD$45.28 in the post group. Patients’ length of stay decreased among female patients by 6.62 days between groups (*p* = 0.049). No other subgroups reached statistical significance for LOS. The full analysis has been included in Appendix C.

### 3.4. Negative PCRNS Outcomes

A subgroup analysis was conducted in those patients with a negative PCRNS in the pre versus post groups, as they represented the patients eligible for intervention and de-escalation (Table 3). In these patients, DOT was further decreased from 96.2 h to 56.5 h in the post group (*p* = 0.014). Additional cost reduction was also noted [USD$79.43 versus USD$50.09] (*p* < 0.01). LOS was numerically longer in the post group, though not statistically significant (*p* = 0.425). Both the pre and post groups had two patients each requiring readmission for a RTI within 30 days (*p* = 0.445).

### 3.5. Regression Analysis

A regression analysis was completed between the continuous variables, anti-MRSA DOT and CCI, to assess for potential confounders of the primary outcome. The pre group did have a numerically higher CCI than the post group, though it was not statistically significant, and both groups were classified as severe on the grading scale. The regression analysis demonstrated that, as CCI increased, the DOT also increased by 5.19 h per one-point increase in the CCI (Figure 3). This was statistically significant (*p* = 0.02) though the association was weak as only 8.3% of the variance was accounted for by the model (R^2^ = 8.3%).

## 4. Discussion

Prior practice at our institution allowed physicians to order PCRNS at their discretion and there was no consistency in practice on the utility of such tool. PCRNS were often ordered through select admission order sets to determine the need for enhanced precautions or MRSA decolonization. Before this study, anti-MRSA agent DOT was determined by providers’ discretion and culture data, if available, to de-escalate therapy. Due to the infrequency of definitive cultures in RTI, the ASP-PCRNS bundle was implemented at our institution to aid clinicians in de-escalation. As such, the pre group intervention would be considered comparing PCRNS alone versus ASP-PCRNS bundle.

The incorporation of the ASP-PCRNS education bundle in our study resulted in a decrease in average DOT by 30.3 h and cost savings of USD$23.61 per RTI patient in a span of three months, which was consistently lower among all demographic subgroups. This decrease in DOT was larger than expected despite logistical hurdles with laboratory results for the PCRNS. Functionality of the PCRNS was likely unaffected by anti-MRSA therapy as the swabs are effective for detection of MRSA in the nares despite anti-MRSA therapy for 48 or more hours [28]. There was no significant worsening of patient outcomes in the in-hospital mortality rate, 30-day readmission, or LOS, and need to restart anti-MRSA therapy was similar between groups. The demographic subgroup analyses showed that females had a significantly lower LOS after ASP-PCRNS bundle implementation. However, due to the further reduction in sample size, the significance of this finding is difficult to ascertain, and larger studies are needed. In our patient population, CCI was correlated with the DOT. However, due to the weak correlation of less than 10%, the effect of this relationship on the ASP-bundle appears to be limited.

In patients who were eligible for intervention based on negative ASP-PCRNS results after bundle implementation noted a reduction of DOT by 39.7 h. Additionally, there was also a significant decrease in cost demonstrated in these patients of USD$29.34 (USD$79.43 vs USD$50.09) (*p* < 0.01). By focusing primarily on negative tests, we can see the impact of the education component of the bundle. Through joint efforts, there were significant reductions in our primary outcomes and secondary outcomes with no increased risks. As compared to the full patient population there was similarly no significant difference in LOS, restart of anti-MRSA therapy, mortality, or readmission between the pre- and postintervention groups.

Currently, four major studies implemented the utility of a pharmacy led ASP-PCRNS protocol to help reduce DOT for anti-MRSA agents. All four studies demonstrated significant reduction in DOT after protocol implementation [21,22,23,24]. Baby et al. found MRSA PCRNS reduced the mean durations of MRSA-targeted therapy by 46.6 h (*p* < 0.0001) and fewer patients required vancomycin serum levels and dose adjustment after implementation (48.1% versus 16.7%; *p* = 0.02). While Baby et al. confirmed that MRSA PCRNS may be used as a diagnostic tool to reduce the DOT, this study had a small sample size and did not have any positive PCRNS patients [21]. Another study, by Willis et al., specifically evaluated patients with pneumonia or acute exacerbation of chronic obstructive pulmonary disease (AECOPD). Patients had a median 2.1-day reduction in vancomycin DOT (2.1 versus 4.2, *p* < 0.0001) with no difference in hospital mortality or LOS. The addition of patients with either pneumonia or AECOPD may be difficult to apply at other institutions since AECOPD therapy does not include MRSA antibiotics [22]. Similarly, Dunaway et al. found median reduction in duration of vancomycin therapy with a difference of 31 h per patient without increasing adverse outcomes (*p* < 0.001). Unlike previous studies the Dunaway trial allowed the final decision on discontinuation of therapy to be at the discretion of the pharmacist and not the provider, but this may not be feasible in other facilities [23]. Dadzie et al. showed there was a significant DOT decrease of 1.4 days and a LOS decrease of 2.8 days after PCRNS implementation with no significant differences in AKI or mortality [24].

Similarly to the above four trials, our study showed that an ASP bundle is safe and effective for usage in RTI patients, despite different methodology in implementation and protocol. Specifically, at our institution we joined efforts with physicians to implement the ASP-PCRNS protocol over a month-long period in an attempt to decrease exposure to anti-MRSA agents. While granting pharmacists the autonomy to discontinue therapy, as the Dunaway et al. trial did, could have further decreased DOT, the scope of practice for pharmacists at our institution would have required modification that was not feasible.

Compared to other studies, Willis et al. was the only study that reported compliance as a measure. They observed similar compliance with PCRNS ordering (79%) as in our study (81%), and overall poor compliance (55%) with de-escalation in those with a negative PCRNS result [22]. At our institution, compliance with the bundle and protocol, assessed at 36 h due to unavoidable delays in PCRNS results, was approximately 10% higher than seen in the Willis et al. study [22].

The limitations of the study include delay in laboratory results, the need for physician review of PCRNS results, and a small sample size that did not meet power. This study was a single-center, retrospective model which may affect the generalization of the study. The delay in results averaging 20 h from order to result time was a rate-limiting step to discontinuation of antibiotics. Our institution had delays in ordering of PCRNS along with the need to send PCRNS for analysis at a centralized system laboratory that only reported results three times a day. Further cost savings and decrease in DOT could have been seen if MRSA PCRNS results were available more rapidly by performing the test at the on-site laboratory. Pharmacists at the institution required an order from physicians to discontinue anti-MRSA therapy, unlike in the Dunaway et al. study which allowed pharmacists to discontinue therapy to their own discretion without an order from a physician [23]. This study did not assess the incidence of acute kidney injury (AKI) or the incidence of new renal replacement therapy initiation due to difficulty in assessing causation as a result of the retrospective nature of the study. Although this was not assessed by the study we expect that our institution has a similar incidence of vancomycin-induced AKI, with a 4–12% increase per day of vancomycin therapy [29,30,31]. However, it is possible that the post group could have had a decreased incidence of AKI as previous literature concluded 48–96 h of therapy was likely insufficient to cause AKI, however further studies are needed to definitely conclude this [8,32]. Cost savings to institutions may be higher than our study describes as we did not account for the reductions of AKI and renal replacement therapy.

We also did not evaluate patients who were prescribed vancomycin for “sepsis” due to pulmonary tract infections. Instead, patients were identified using an in-house antibiotic tracking system that relied upon physicians to correctly select the appropriate indication upon ordering of anti-MRSA therapy. Use of this tracking system may have resulted in the loss of potentially eligible patients if the physicians did not select the appropriate indication that could have been identified had ICD-10 codes been used to develop the patient list instead. However, due to the institution-approved process of ordering a PCRNS specifically in patients with a documented RTI on anti-MRSA therapy, an ICD-10 patient list would not have been an accurate representation of the bundle results due to exclusion from the standardized workflow. These factors along with the increased education on anti-MRSA therapy for RTI could have contributed to the failure of the study to include enough patients to adequately power the study.

## 5. Conclusions

The implementation of a multi-disciplinary ASP-PCRNS protocol decreased vancomycin DOT and reduced cost. These results demonstrated PCRNS can be a useful tool to guide clinicians to discontinue unnecessary therapy without negatively impacting patients. Further studies utilizing a similar multidisciplinary approach that have a larger sample size, longer study period, multicenter approach, and which include cost avoidance outcomes would be useful.

## Figures and Tables

**Figure 1 pharmacy-08-00101-f001:**
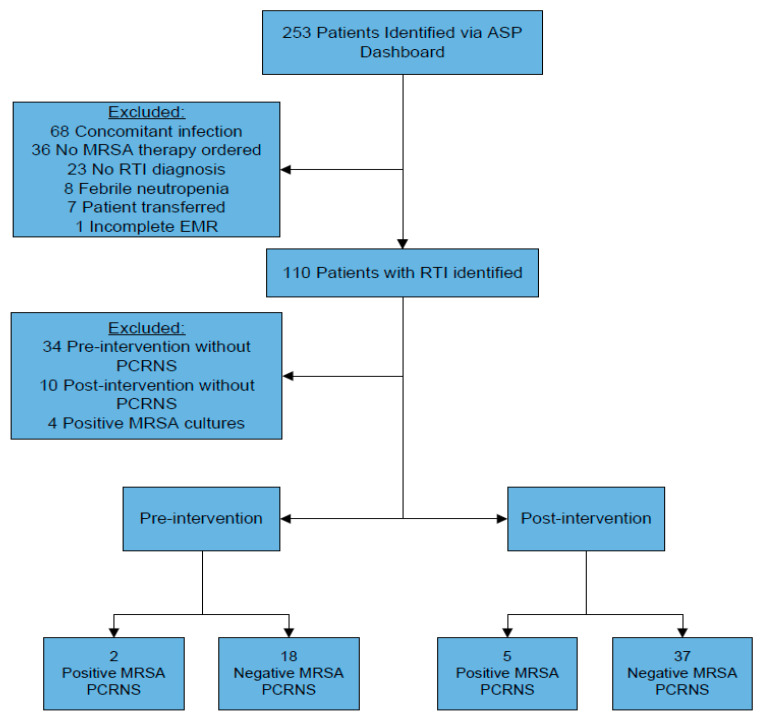
Screening flow diagram of patients in the pre- and postintervention groups.

**Figure 2 pharmacy-08-00101-f002:**
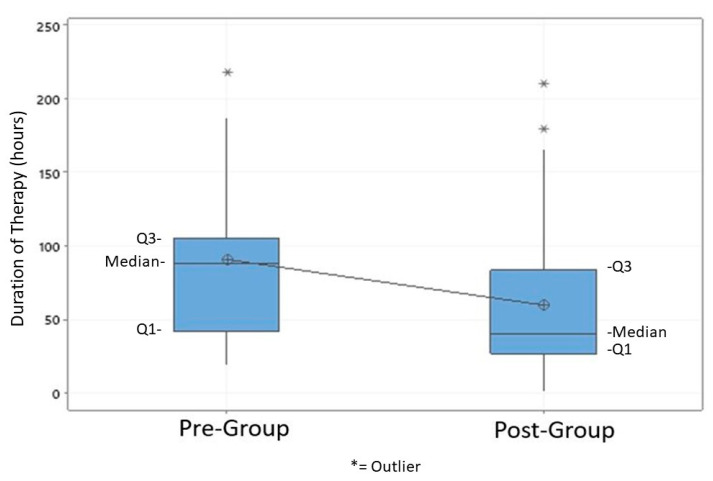
Overall duration of therapy between patients in the pre- and postintervention groups.

**Figure 3 pharmacy-08-00101-f003:**
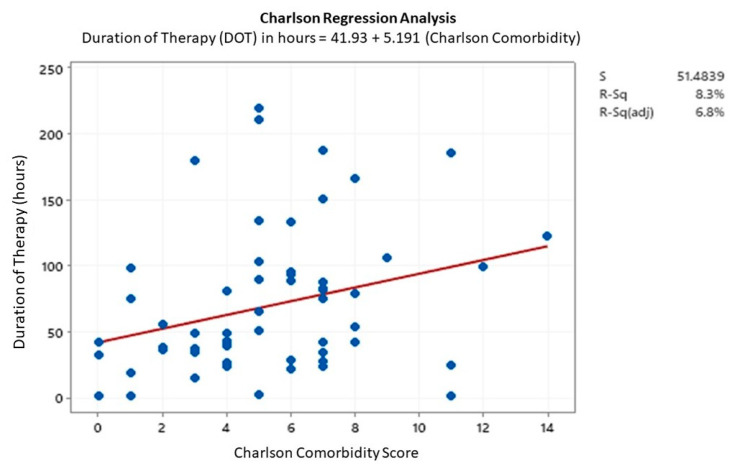
Regression analysis model comparing the relation between the Charlson Comorbidity Index score and duration of therapy.

**Table 1 pharmacy-08-00101-t001:** Patient baseline characteristics in the preintervention and postintervention groups.

Baseline Characteristics	Pre Group (*n* = 20)	Post Group (*n* = 42)	*p*-Value
Male, *n* (%)	10 (50)	24 (57.1)	0.597
Female, *n* (%)	10 (50)	18 (42.8)	
Age, mean (std)	65.6 (19.9)	66.9 (23.0)	0.827
Caucasian, *n* (%)	13 (65)	26 (61.9)	0.78
Asian, *n* (%)	1 (5)	1 (2.4)	
Black or African American, *n* (%)	6 (30)	13 (31.0)	
More than one race, *n* (%)	0	1 (2.4)	
Unknown/not reported, *n* (%)	0	1 (2.4)	
BMI, mean (std)	26.4 (6.2)	26.4 (6.3)	0.983
Charlson Comorbidity Index, mean (std)	6.1 (3.5)	5.0 (2.7)	0.256
Creatinine Clearance at Anti-MRSA start, mean (std)	72.4 (39.8)	66.7 (28.2)	0.573
Concomitant Nephrotoxin Usage, *n* (%)	20 (100)	41 (97.6)	−
Negative PCR Nasal Swabs, *n* (%)	18 (90)	37 (88.1)	0.824

**Table 2 pharmacy-08-00101-t002:** Primary and secondary outcomes of the pre and post intervention groups*.

PCRNS Endpoints	Pre Group (*n* = 20)	Post Group (*n* = 42)	*p*-Value
Duration of therapy in hours, mean (std)	90.7 (54.4)	59.7 (50.4)	0.039
Hospital length of stay in days, mean (std)	10.9 (6.2)	17.3 (48.9)	0.407
Anti-MRSA therapy restarted, *n* (%)	1 (5)	2 (4.7)	0.967
In-hospital mortality, *n* (%)	2 (10)	4 (9.5)	0.953
30-day readmission, *n* (%)	2 (10)	2 (4.7)	0.432
Cost in US dollars, median (IQR)	75.3 (51.7–95.4)	51.7 (34.1–67.2)	<0.01

* Primary and secondary outcomes include patients with both positive and negative polymerase-chain-reaction nasal swabs (PCRNS) results in the pre- and postintervention groups.

**Table 3 pharmacy-08-00101-t003:** Outcomes in the pre- and postintervention groups for patients with negative PCRNS.

Negative PCRNS Endpoints	Pre Group (*n* = 18)	Post Group (*n* = 37)	*p*-Value
Duration of therapy in hours, mean (std)	96.2 (54.7)	56.5 (47.7)	0.014
Cost in US dollars, median (IQR)	79.4 (55.2–96.1)	50.1 (33.5–64.2)	<0.01
Length of stay, mean (std)	11.1 (6.52)	18.1 (52.1)	0.425

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
