# Peer review of "Impact of Nasal Swabs on Empiric Treatment of Respiratory Tract Infections (INSERT-RTI)"

_pharmacy, 2020, doi:10.3390/pharmacy8020101_

Round 1
Reviewer 1 Report
See attach document

Author Response
Major Comments
- Materials and methods:
- The material and methods need to be re-written for clarification. It is written in narrative style, but the purpose of Material and methods is to write clear protocols that allow other researchers to perform a similar study in order to compare de results. It needs to be divided into clear sections with well written guidelines describing the work performed.
- Materials and methods was divided into clear sections including overall description of inclusion and exclusion, IRB, and time-period. Then further divided into sections such as educational information, baseline characteristics and statistical analysis to guide the reader through the reading and be able to replicate the study.
- Statistics:
- There is not clear criteria for inclusion of patients in this study, how statistical analysis were performed,… The first paragraph of the results section talks about inclusion and exclusion of patients without previous context.
- The inclusion/exclusion criteria was updated and clearly defined in the material and methods section. A statistical analysis section was added to include further description of statistics performed.
- Results:
- May be broken into sections that each one has a specific message. This will aid the reader to better understand the important findings of this research. The results section will also benefit from briefly discussing the implications of each finding.
- The results section has been divided into clear titled sections to give the reader continuity throughout the read.
- Discussion:
- Starts stating “Recent studies….” But no references are added in this sentence. A little after (line 197) there is a reference to the worked performed by Willis et al, but again there is no reference added to this sentence.
- The start of the discussion has been changed to describe the practice at our institution prior to the study implementation. References describing any studies have been included.
- Starts stating “Recent studies….” But no references are added in this sentence. A little after (line 197) there is a reference to the worked performed by Willis et al, but again there is no reference added to this sentence.
- May be broken into sections that each one has a specific message. This will aid the reader to better understand the important findings of this research. The results section will also benefit from briefly discussing the implications of each finding.
- There is not clear criteria for inclusion of patients in this study, how statistical analysis were performed,… The first paragraph of the results section talks about inclusion and exclusion of patients without previous context.
- The material and methods need to be re-written for clarification. It is written in narrative style, but the purpose of Material and methods is to write clear protocols that allow other researchers to perform a similar study in order to compare de results. It needs to be divided into clear sections with well written guidelines describing the work performed.
Reviewer 2 Report
Methicillin-resistant Staphylococcus aureus (MSRA) is a common cause of lower respiratory tract infections (LRTIs), particularly pneumonia, leading to related morbidity and mortality. Vancomycin is the primary treatment option for infections caused by MSRA. However, it is not without serious side effects, hence vancomycin warrants judicious use to prevent the generation of vancomycin-resistant Staphylococcus aureus and side effects. The present study describes the benefits ASP-PCRNS bundled with education programs in collaboration with health care professionals to reduce anti-MSRA interventions and associated RTI, LOS, hospital readmission, and cost. The findings in the paper support the growing need to reduce antibiotic-based intervention early after hospitalization and the usefulness of PCRNS as a vital tool in concluding to that decision. My comments are as follows,
- The number of patients in the cohort is considerably smaller (the author has mentioned in the text though) in the study to extrapolate conclusions decisively.
- In the materials and method, the authors talk about the requirement of 126 patients for the experiments, to begin with, but 110 patients are enrolled during the actual study, what was the reason to this?
- The hospital length of stay (LOS) has increased in the post-intervention group by 7 days (with very high std deviation) compared to the pre-intervention group. Have authors tried calculating LOS in both groups with negative PCRNS?
Author Response
Reviewer 2
- The number of patients in the cohort is considerably smaller (the author has mentioned in the text though) in the study to extrapolate conclusions decisively.
- The study sample was smaller than anticipated with a higher than expected exclusion rate. Unfortunately due to time constraints as this was PGY1 residency project and a pilot for our health system we are unable to expand the study beyond the current time frame. Thank you for your feedback.
- In the materials and method, the authors talk about the requirement of 126 patients for the experiments, to begin with, but 110 patients are enrolled during the actual study, what was the reason to this?
- The sample size of this study represents the amount of patients with a respiratory tract infection that met inclusion. The reasons for exclusion are included in Figure 1 (191/253). Additionally, the education provided caused more providers to evaluate the need for vancomycin in patients without risk factors which results in less patients started on anti-MRSA therapy.
- The hospital length of stay (LOS) has increased in the post-intervention group by 7 days (with very high std deviation) compared to the pre-intervention group. Have authors tried calculating LOS in both groups with negative PCRNS?
- Updated to include the LOS within the PCRNS noted in Table 3.
Reviewer 3 Report
Huffman et al. study the use of MRSA diagnostic PCRNS combined with an education bundle to lower the treatment cost significantly and an informed decision based on negative predictive value of PCRNS in reducing the DOT. Author addresses that PCRNS alone has been recently used and studies addressing that to reduce the DOT were nicely reviewed in the discussion part. However, since author proposes a MRSA with education bundle and claim is that it not only reduced the DOT but also lowered the cost associated with treatments.
I have following concerns that author needs to address:
- There is no direct comparison of PCRNS alone against PCRNS with education bundle. How author believe that the conclusion supports the use of bundle over PCRNS alone.
- I do not understand how the ASP-PCRNS bundle is multi-disciplinary protocol. The ASP educational package’s key elements should be provided as supplementary data.
- Following description needs to be simplified as it is very confusing: In the discussion at line 258, This may have resulted in patients who were eligible for inclusion …. Used to develop the patient list.
Author Response
Reviewer 3
- There is no direct comparison of PCRNS alone against PCRNS with education bundle. How author believe that the conclusion supports the use of bundle over PCRNS alone.
- Prior practice at our institution allowed physicians to order PCRNS to their discretion but there was no consistency in practice among providers and pharmacists on the utility of PCRNS. While PCRNS were available for several years and is often used during admission to help identify patients colonized with MRSA to administer mupirocin, PCRNS were not a standard of care. As such, the pre-group intervention would be considered comparing PCRNS alone versus ASP-PCRNS bundle. We have updated the discussion to further clarify this notion.
- I do not understand how the ASP-PCRNS bundle is multi-disciplinary protocol. The ASP educational package’s key elements should be provided as supplementary data.
- ASP educational material has been included in the supplemental appendix to guide others in adapting the protocol into their institutions. Pharmacists, medical residents, and attending physicians joined to educate colleagues via a variety of methods.
- Following description needs to be simplified as it is very confusing: In the discussion at line 258, This may have resulted in patients who were eligible for inclusion …. Used to develop the patient list.
- This sentence has been updated. Patients were identified using an in-house antibiotic tracking system that relied upon physicians to correctly select the appropriate indication upon ordering of anti-MRSA therapy. This may have resulted in the loss of potentially eligible patients if the physicians did not select the appropriate indication within the antibiotic order that could have potentially been identified had ICD-10 codes been used to develop the patient list instead.
Reviewer 4 Report
This paper attempts to expand upon current literature demonstrating rapid PCR MRSA nasal swabs as an effective de-escalation strategy for anti-MRSA agents. The study was conducted over 6 months at a community hospital and unfortunately did not meet power but showed a statistically significant difference in DOT. Comments for consideration are listed below:
- I'm curious why the timeframe was done so short? Three months pre and post in this instance ended up causing the study to not meet power because of high number of exclusions. A longer study would likely capture better the overall effectiveness/adherence of the bundle. Additionally, due to the seasonal nature of RTI, going at least a year would best account for true benefit. Ideally this should be done for a longer timeframe.
- How were costs of various drugs/tests determined? There is no reference/explanation for costs given.
- Since there were no linezolid patients in study, I would just state as much in methods/results and not mention linezolid anywhere else such including abstract and conclusion.
- Was the ASP dashboard acted upon "real-time"? Who actually evaluated this and acted on it? How often was it reviewed if not real-time?
- In the pre-group, what % of patients were de-escalated from negative MRSA nares within 36 hours? Only the post-group is reported so a comparison would be interesting and notable.
- Lines 180-186. This discussion of 7 patients including table 4 doesn't really add a lot to manuscript and difficult to make any meaningful conclusion on 7 patients.
- The year long cost savings should not be extrapolated from 3 months of data due to seasonal variation of pneumonia. Would delete this.
- There is recent data demonstrating that even one more day of vancomycin therapy (prophylaxis data) may increase risk of AKI. With the 30 hour difference in vancomycin DOT, I think it is worth evaluating incidence of AKI despite the author's concerns over definitive attribution of possible less AKI to de-escalation intervention. The Charlson comorbidity index is similar between groups as well as receipt of nephrotoxic agents.
Author Response
Reviewer 4
- I'm curious why the timeframe was done so short? Three months pre and post in this instance ended up causing the study to not meet power because of high number of exclusions. A longer study would likely capture better the overall effectiveness/adherence of the bundle. Additionally, due to the seasonal nature of RTI, going at least a year would best account for true benefit. Ideally this should be done for a longer timeframe.
- The study was conducted in a 7 month timespan due to time constraints as this was PGY1 residency project and a pilot project for our health system. We are unable to expand the study beyond the current time frame due to these factors. Thank you for your feedback.
- How were costs of various drugs/tests determined? There is no reference/explanation for costs given.
- Cost was broken down and described under baseline characteristics. Laboratory costs were provided by the institution laboratory. Drug costs were determined by 3rd party resource such as Lexicomp for each agent.
- Since there were no linezolid patients in study, I would just state as much in methods/results and not mention linezolid anywhere else such including abstract and conclusion.
- Linezolid was deleted and will no longer be mentioned in the abstract and conclusion.
- Was the ASP dashboard acted upon "real-time"? Who actually evaluated this and acted on it? How often was it reviewed if not real-time?
- The dashboard allowed the collection of EMR from current and prior months. The ASP committee, which is led by the infectious disease pharmacist, reviews the dashboard regularly. This description was included in the materials and methods section. “A patient list was generated using an in-house antibiotic tracking system that recorded antimicrobial agents and their indications. Patients identified on vancomycin or linezolid with RTI as the only documented indication by the tracking system were then screened for inclusion in the study.
- In the pre-group, what % of patients were de-escalated from negative MRSA nares within 36 hours? Only the post-group is reported so a comparison would be interesting and notable.
- 22% of the pre-group were acted upon with 36 hours of PCRNS.
- Lines 180-186. This discussion of 7 patients including table 4 doesn't really add a lot to manuscript and difficult to make any meaningful conclusion on 7 patients.
- Thank you for the feedback. We have removed this section from the manuscript.
- The year long cost savings should not be extrapolated from 3 months of data due to seasonal variation of pneumonia. Would delete this.
- Thank you for the feedback. This information was deleted from the manuscript “When extrapolated out to 12 months of data our program could result in a cost savings for the institution of approximately USD$4,000 per year though this only accounts for directly tangible costs of medications and laboratory testing while missing intangible costs.”
- There is recent data demonstrating that even one more day of vancomycin therapy (prophylaxis data) may increase risk of AKI. With the 30 hour difference in vancomycin DOT, I think it is worth evaluating incidence of AKI despite the author's concerns over definitive attribution of possible less AKI to de-escalation intervention. The Charlson comorbidity index is similar between groups as well as receipt of nephrotoxic agents.
- AKI is unfortunately unable to be assessed at this time due to a lack of data collection that is required to assess the outcome.
Reviewer 5 Report
Thank you for allowing me to review this article. Huffman et al sought to evaluate the impact of a bundled intervention (nasal swabs and physician education) on anti-MRSA treatment durations in adults with respiratory tract infections. While this multidisciplinary team of a pharmacist, emergency medicine physician and resident is different, the bundled intervention itself is not and has been well described in the stewardship literature. The major flaw with this submission is the methods (see below).
Consider shortening the introduction and moving the review of prior studies to the discussion portion of this manuscript. The stewardship argument for de-escalation from MRSA-active antibiotics is not that vancomycin and linezolid are broad-spectrum. Conversely, it is more that patients do not need unnecessary MRSA-active antibiotic exposure if they are not colonized with MRSA, as demonstrated by the negative PCR nasal swabs. Thus, would revise the argument in lines 37-40.
The methods lack a regression analysis to independently conclude that the education bundle reduced vancomycin treatment duration. In the absence of a regression analysis, it is invalid to make a meaningful conclusion and it would only be the crude comparisons that were made. I would recommend that the authors consider an interrupted time series as a more robust method and also include more data (i.e., 6 months post-data). The inclusion criteria are also flawed and should only be patients with negative nasal swabs who were initiated on vancomycin for pneumonia. While comprising 10% of the population in each group (which is large), it is unclear how the positive nasal swab patients were analyzed in this study--were they excluded from the primary analysis? Furthermore, a quarter (25%) of the patients did not receive PCR nasal swabs within 36 hours of vancomycin initiation. It is concerning that vancomycin was initiated, and then PCR nasal swab was obtained because systemic vancomycin exposure >36 hours may significantly reduce the swab's sensitivity. It would not be good practice to make de-escalation recommendations based off nasal swabs alone unless respiratory cultures are also obtained and indicate that there is no MRSA present.
Overall, I commend the authors for implementing this program at their institution. A closer evaluation of that subgroup of patients who received PCR nasal swabs >36 hours of vancomycin initiation should be conducted.
Author Response
Reviewer 5
- Consider shortening the introduction and moving the review of prior studies to the discussion portion of this manuscript. The stewardship argument for de-escalation from MRSA-active antibiotics is not that vancomycin and linezolid are broad-spectrum. Conversely, it is more that patients do not need unnecessary MRSA-active antibiotic exposure if they are not colonized with MRSA, as demonstrated by the negative PCR nasal swabs. Thus, would revise the argument in lines 37-40.
- Introduction was condensed and elaborated on risk of resistance and side effects without any gained benefit. Review of prior studies was moved to the discussion.
- The methods lack a regression analysis to independently conclude that the education bundle reduced vancomycin treatment duration. In the absence of a regression analysis, it is invalid to make a meaningful conclusion and it would only be the crude comparisons that were made. I would recommend that the authors consider an interrupted time series as a more robust method and also include more data (i.e., 6 months post-data).
- Thank you for your review. We have consulted our statistician and currently we don’t have sufficient data to run a segmented regression to conduct a time series analysis. This is a study pilot conducted over a short period of time, intended to be completed from implementation to analysis within a 10-month period. We will be conducting a time series analysis along with other appropriate analyses when this study pilot is implemented to the other adult hospitals within our healthcare system. At this time we have added a regression analysis to evaluate the association of Charlson Comorbidity index (CCI) and duration of therapy within our study period. The regression shows a statistically significant association between CCI and time to discontinuation of vancomycin. The regression analysis demonstrated that as CCI increased the time to discontinuation of anti-MRSA therapy increased.
- The inclusion criteria are also flawed and should only be patients with negative nasal swabs who were initiated on vancomycin for pneumonia.
- Patients were included whether they had a positive or negative nasal swab. A subgroup analysis was further done to identify the results in patients with negative results.
- While comprising 10% of the population in each group (which is large), it is unclear how the positive nasal swab patients were analyzed in this study--were they excluded from the primary analysis?
- The primary analysis included patients with a positive and negative PCRNS. The primary outcomes measured whether a PCRNS would help reduced the duration of therapy. The subgroup analysis further analyzed the effect from a negative PCRNS result. Positive PCRNS results were not further analyzed.
- Furthermore, a quarter (25%) of the patients did not receive PCR nasal swabs within 36 hours of vancomycin initiation. It is concerning that vancomycin was initiated, and then PCR nasal swab was obtained because systemic vancomycin exposure >36 hours may significantly reduce the swab's sensitivity. It would not be good practice to make de-escalation recommendations based off nasal swabs alone unless respiratory cultures are also obtained and indicate that there is no MRSA present.
- Studies have identified PCRNS to have high sensitivity, specificity, and negative predictive value (NPV) comparing the results with cultures. As such, we did not compare the efficacy of the swab. We excluded patients with positive MRSA culture results. This manuscript addressed the studies identifying PCRNS reliability. PCRNS have been shown to remain effective for detection of MRSA in the nares despite vancomycin therapy of 48 hours or greater. See comment on Shenoy et al. in discussion regarding accuracy of PCRNS despite prolonged vancomycin therapy.
- Overall, I commend the authors for implementing this program at their institution. A closer evaluation of that subgroup of patients who received PCR nasal swabs >36 hours of vancomycin initiation should be conducted.
- PCRNS have been shown to remain effective for detection of MRSA in the nares despite vancomycin therapy of 48 hours or greater.
Round 2
Reviewer 1 Report
Impact of Nasal Swabs on Empiric Treatment of Respiratory Tract Infections (INSERT-RTI) This manuscript provides a detail study about the efficacy of nasal swabs on the empiric treatment of respiratory infections, while evaluating the costs of treatment and the outcome of the patients after the intervention. Major comments: The revised versions of this manuscripts has substantially improved offering a detailed description of the material and methods as well as inclusion and exclusion criteria. 1. There is no description of the timeline. When this study was perform? How long it lasted for? It is important to know the time of the year as well as the length as these data are important for the evaluation of the efficacy. 2. Table 1 will benefit from having an extra-row of "Female". If possible also identify the raze of the non-caucasian patients. Is it any information available regarding other risks factors? for example, is any of the patients recipient of a transplant of cystic fibrosis patients, etc.? 3. Table 2 is assessing the duration of therapy, hospital length and other parameters as pre-groups and post-group. Could a new table be included evaluating these parameters in relation with gender, age, race and BMI? This analysis might not appear to be important but it has high significance as this test can be more valuable in certain type of patients than others. Also, this analysis will allow to better understand why some patients respond to this better than others. 4. The discussion is nicely written and well thought, nevertheless, it can be substantially improved after including these analysis that can enlighten the interpretation of the data. Overall, it is a nice manuscript that is improved from the prior version, however, it still needs more work to better understand the data produced during this study.Author Response
Reviewer 1:
1. There is no description of the timeline. When this study was perform? How long it lasted for?
The timeline is described in lines 71 and 72 of the manuscript. The pre-intervention and post-intervention arms were between June 1st 2019 and August 30th 2019, and October 1st 2019 to January 1st 2020, respectively. The month of September 2019 was a washout period to allow for education and process implementation.
2. Table 1 will benefit from having an extra-row of "Female".
A row with female gender has been added to the gender category under the baseline characteristics table 1.
If possible also identify the race of the non-Caucasian patients.
Additional information on race for both groups have been added to the baseline characteristics table 1.
Baseline Characteristics |
Pre-Group (n=20) |
Post-Group (n=42) |
P-value |
Male, n (%) |
10 (50) |
24 (57.1) |
0.597 |
Female, n (%) |
10 (50) |
18 (42.8) |
|
Age, mean (std) |
65.6 (19.9) |
66.9 (23.0) |
0.827 |
Caucasian, n (%) |
13 (65) |
26 (61.9) |
0.78 |
Asian, n (%) |
1 (5) |
1 (2.4) |
|
Black or African American, n (%) |
6 (30) |
13 (31.0) |
|
More than one race, n (%) |
0 |
1 (2.4) |
|
Unknown/not reported, n (%) |
0 |
1 (2.4) |
|
BMI, mean (std) |
26.4 (6.2) |
26.4 (6.3) |
0.983 |
Charlson Comorbidity Index, mean (std) |
6.1 (3.5) |
5.0 (2.7) |
0.256 |
Creatinine Clearance at Anti-MRSA start, mean (std) |
72.4 (39.8) |
66.7 (28.2) |
0.573 |
Concomitant Nephrotoxin Usage, n (%) |
20 (100) |
41 (97.6) |
- |
Negative PCR Nasal Swabs, n (%) |
18 (90) |
37 (88.1) |
0.824 |
Is it any information available regarding other risks factors? For example, is any of the patients recipient of a transplant of cystic fibrosis patients, etc.?
Our study includes the Charlson Comorbidity index (CCI), which is a standardized validated tool to predict a 1-year mortality risk based on a composite score of comorbidities in categories including cardiovascular, pulmonary, neurologic, renal, hepatic, gastrointestinal, and neoplastic. The scores are categorized into three grades: mild scores (1-2), moderate scores (3-4), or severe (≥ 5).
We did not look at specific risk factors for multi-drug resistant organisms as the study focused solely on MRSA and all patients were required to have an MRSA PCRNS which was a definitive indicator for the risk of MRSA infection. Additionally, within our healthcare system a separate hospital serves as the primary cystic fibrosis center so exposure to these patients at our institution is extremely rare. We unfortunately do not have any data regarding patients’ receipt of transplant or concomitant immunosuppressive agents outside of use captured within the CCI.
3. Table 2 is assessing the duration of therapy, hospital length and other parameters as pre-groups and post-group. Could a new table be included evaluating these parameters in relation with gender, age, race and BMI?
A table evaluating duration of therapy, length of stay and cost was generated comparing specific baseline characteristics. For certain analyses, the method of comparison was changed from a t-test to the Mann-Whitney U test due to either small comparator groups or non-normal populations.
Among the whole population both cost and duration of therapy were significant, but not all subgroups met significance which could potentially be due to the demographic in which they were sorted or due to the inability to power the statistical test to determine significance. Duration of therapy was statistically significantly shorter in the post group among female patients, non-white patients, and patients with BMIs less than 30. All other parameters did not meet statistical significance though each population had a numerically lower duration of therapy in the post group than in the pre group. Likewise, cost was overall reduced among all groups in the post group compared to those in the pre group. However only in patients with a BMI less than 30 kg/m2 had statistically significant reduction in cost, USD$80.62 in the pre group versus USD$45.28 in the post group. Patient’s length of stay decreased in female patients by 6.62 days in the post group as compared to the pre group which was statistically significant. It is difficult to assess causation in the subgroup analyses, however in all endpoints tested females had either a numerically or statistically larger difference between the pre and post groups as compared to males so further studies with a larger sample size are needed to further investigate and assess the impact of gender.
4. The discussion can be substantially improved after including these analysis that can enlighten the interpretation of the data.
A table analyzing duration of therapy, length of stay and cost has been generated comparing baseline characteristics. Data has been summarized in the results section and described in the discussion. The full table of analyses added to appendix 3.
Baseline Characteristics |
Pre-Group (n=20) |
Post-Group (n=42) |
P-value |
|
|
|
|
Duration of therapy |
|
|
|
Gender |
Mean (Std Dev)/ Median** |
Mean (Std Dev)/ Median** |
|
Male |
67.9 (30.6) |
57.8 (47.5) |
0.470 |
Female |
94.1** |
35.6** |
0.047 |
Age |
Mean (Std Dev)/ Median** |
Mean (Std Dev)/ Median** |
|
< 65 years old |
65.8** |
36.6** |
0.057 |
> 65 years old |
106.1(62) |
74.8 (58) |
0.158 |
Race |
Mean (Std Dev)/ Median** |
Mean (Std Dev)/ Median** |
|
Caucasian |
81.7 (42.8) |
67.8 (53.9) |
0.389 |
Non-Caucasian |
98.5** |
34.3** |
0.035 |
BMI |
Mean (Std Dev)/ Median** |
Mean (Std Dev)/ Median** |
|
< 30 kg/m2 |
84.2** |
37.2** |
0.031 |
> 30 kg/m2 |
89.1** |
74.6** |
0.316 |
Length of Stay |
|
|
|
Gender |
Mean (Std Dev)/ Median** |
Mean (Std Dev)/ Median** |
|
Male |
7.8** |
8.5 |
0.610 |
Female |
14.9 (7.0) |
8.3 (7.0) |
0.049 |
Age |
Median** |
Median** |
|
< 65 years old |
8.0** |
3.2** |
0.112 |
> 65 years old |
9.3** |
13.3** |
0.987 |
Race |
Mean (Std Dev)/ Median** |
Mean (Std Dev)/ Median** |
|
Caucasian |
11.4 (7.2) |
10.3 (7.1) |
0.660 |
Non-Caucasian |
9.4** |
4.1** |
0.193 |
BMI |
Mean (Std Dev)/ Median** |
Mean (Std Dev)/ Median** |
|
< 30 kg/m2 |
8.3* |
5.3* |
0.274 |
> 30 kg/m2 |
10.2* |
13.6* |
0.953 |
Cost |
|
|
|
Gender |
Median (IQR) |
Median (IQR) |
|
Male |
59.8 |
47.2 |
0.108 |
Female |
84.5 |
46.3 |
0.052 |
Age |
Median |
Median |
|
< 65 years old |
57.7 |
46.6 |
0.066 |
> 65 years old |
82.7 |
53.3 |
0.089 |
Race |
Median** |
Median** |
|
Caucasian |
79.2** |
48.4** |
0.092 |
Non-Caucasian |
69.9** |
45.9** |
0.057 |
BMI |
Median** (IQR) |
Median* (IQR) |
|
< 30 kg/m2 |
80.6 (48.6-101.9) |
45.3 (30.3-64.6) |
0.023 |
> 30 kg/m2 |
74.6** |
63.6** |
0.444 |
* Mann-Whitney-U
Reviewer 4 Report
The authors addressed all of my concerns with exception of one in the revised manuscript. They detailed in their response regarding the dashboard that it was reviewed regularly by the ASP committee, which is led by the ID pharmacist. If they could add this into the paper and quantify "regularly" as far as how often it was reviewed (i.e. once daily) and when it was reviewed (Monday-Friday or everyday including weekends) that would be helpful.
Author Response
Reviewer 4:
If they could add details on the ASP Dashboard into the paper and quantify "regularly" as far as how often it was reviewed (i.e. once daily) and when it was reviewed (Monday-Friday or everyday including weekends).
- A prospective patient list was generated and reviewed daily including weekends using an in-house antibiotic tracking system that recorded antimicrobial agents and their indications. Found in line 78 of the manuscript.
Round 3
Reviewer 1 Report
I would like to congratulate the authors for the revised version of the manuscript. It has improved considerably and I believe it is a great preliminary study that provides important data for the scientific community and particularly the medical field.